# Association between Periodontal Disease and Levels of Triglyceride and Total Cholesterol among Korean Adults

**DOI:** 10.3390/healthcare8030337

**Published:** 2020-09-12

**Authors:** Seon-Rye Kim, Seoul-Hee Nam

**Affiliations:** 1Department of Pharmacy, College of Pharmacy, Kangwon National University, Chuncheon-si 24341, Korea; sjsanj@hanmail.net; 2Department of Dental Hygiene, College of Health Science, Kangwon National University, Samcheok-si 25945, Korea

**Keywords:** periodontal disease, triglyceride, dyslipidemia, total cholesterol

## Abstract

Although the correlation between periodontal condition and hyperlipidemia was shown by previous researches, it remains disputed. This study was based on data from the seventh Korea National Health and Nutrition Examination Survey 2016–2018. Data included 14,068 subjects’ demographic characteristics, total cholesterol levels, triglyceride levels, and periodontal conditions. We analyzed the correlation between periodontal disease and blood lipid levels using complex-sample chi square tests and complex-sample logistic regression. The results of chi square tests showed a significant difference in demographic characteristics according to total cholesterol level and triglyceride level. The results of logistic regression, adjusted for the subjects’ demographic characteristics such as age, gender, house income, marital status, home ownership, number of persons living together, health insurance coverage, and economic activity, showed that prevalence of periodontal disease was 1.048-fold (confidence interval (CI) 0.827–1.133) higher in the high-total-cholesterol group and 1.499-fold (CI 1.576–1.983) higher in high-triglyceride group. We found that not high total cholesterol but high triglyceride level was connected with periodontal disease. Therefore, management of triglyceride level could be a possible precaution of periodontal disease.

## 1. Introduction

With increasing life expectancy, the importance of major oral diseases that threaten oral health has been emphasized more of late [1]. As the desire for good health has increased, the interest in oral health, an essential element of health, has also increased [2]. Periodontal disease occurs in the surrounding tissues supporting the teeth, and most periodontal diseases take the form of chronic inflammatory periodontitis that progresses slowly but progresses and stops repeatedly depending on the individual’s systemic health and oral care status [3]. Periodontal disease is a major oral health problem and causes loss of teeth by destroying the connective tissue and bones that support the teeth [4].

Richmond et al. [5] reported that neglected oral healthcare adversely affects general health. As oral disease does not directly threaten life, it is easy to neglect oral healthcare. As a result, various problems occur, such as poor quality of life, risk factors for chronic diseases like heart disease and diabetes, indigestion, and problems with interpersonal relationships [6].

Cardiovascular diseases were reported to have a significant effect on periodontal diseases, and South Korean adults with hypertension or cardiovascular disease are more likely to lose their teeth [7].

It was reported that, due to the relationship between periodontal disease and cardiovascular disease, a mediator that causes inflammation in the oral cavity affects the inflammatory response of the periodontal tissue and the whole body through the bloodstream [7,8]. In addition, it was reported that various risk factors, such as health behaviors and daily lifestyle habits, have a common effect on periodontal and cardiovascular diseases [9]. Previous studies reported that cardiovascular disease and periodontal disease are connected to each other. The studies on the relationship between dyslipidemia and periodontal disease, however, are very insufficient. Due to the changes in eating habits and lifestyles with the rapid advance of modern society, the trend of disease occurrence is also changing; thus, it must be continuously managed on the basis of the relationship between systemic and oral diseases. Therefore, much attention and effort are needed to promote oral health.

This study aimed to investigate the causal relationship between periodontal and cardiovascular diseases and to provide fundamental data suggesting the importance of periodontal care by analyzing the impact of the presence or absence of dyslipidemia on the induction of periodontal disease using data from the seventh Korea National Health and Nutrition Examination Survey (KNHANES VII) conducted for 3 years, from 2016 to 2018.

## 2. Materials and Methods 

### 2.1. Study Design and Subjects

The Korea Centers for Disease Control and Prevention (KCDC) performed KNHANES VII from 2016–2018. The KNHANES is made up of nationally representative samples obtained from standard household surveys using a systematic sampling method. The original survey consisted of 24,269 people. This cross-sectional study was restricted to subjects over 19 years of age. The analyses in this study were confined to 14,068 respondents. All participants signed an informed consent form prior to participation. The Institutional Research Ethics Review Committee of KCDC approved this study. KNHANES VII obtained the approval for the third year of the study (2018-01-03-P-A). In the first and second years, KNHANES VII was overseen by the government for public welfare, and the survey was conducted without going through a review by the Research Ethics Review Committee, as a national health nutrition survey may be conducted without deliberation by the Research Ethics Review Committee in accordance with Article 2 subparagraph 1 of the Bioethics Act and Article 2 (2) 1 of the Enforcement Rules of the same Act.

### 2.2. Demographic Characteristics

Demographic variables included gender, age, body mass index (BMI), marital status, education level, home ownership, number of persons living together, health insurance coverage, house income, and economic activity. Age was categorized into four groups: <30, 30–49, 50–69, and ≥70. BMI was categorized into three groups: underweight (<18.5), normal (18.5–24.9), and obese (≥25). Education level was classified into four categories: below elementary school, graduated middle school, graduated high school, and over college. Number of persons living together was classified as alone or >2 persons. Health insurance coverage was categorized into two groups as national health insurance service or medical care service. Household income was classified into five categories: low, low–middle, middle, middle–high, and high. Economic activity was categorized into three groups: none (including students and housewives), office jobs, and nonclerical jobs.

### 2.3. The Periodontal Disease

The KNHANES used the community periodontal index (CPI) to evaluate the presence of periodontal disease. For the CPI invented by the World Health Organization (WHO), the mouth was divided into sextants and each sextant was examined. The CPI was measured with a ball probe used by dentists and the highest score was recorded. When the CPI was 3 or higher, the subjects were considered as patients with periodontal disease [10].

### 2.4. Characteristics of Blood Lipid Levels

In terms of the blood lipid levels, the 12,733 subjects who had total cholesterol level <240 mg/dL were made part of the normal-total-cholesterol group (NTCG), the 1335 subjects who had total cholesterol level >240 mg/dL were made part of the high-total-cholesterol group (HTCG), the 12,070 subjects who had triglyceride level <200 mg/dL were made part of the normal-triglyceride group (NTG), and the 1998 subjects who had triglyceride level >200 mg/dL were made part of the high-triglyceride group (HTG). The diagnostic criteria were based on those of the Korean Society of Lipidology and Atherosclerosis.

### 2.5. Statistical Analysis

The individual weighted factors were used, and a complex sampling design of the survey was considered to obtain the variances. The difference between the demographic variables and blood lipid levels (total cholesterol, triglyceride) was obtained by chi square tests. Complex-sample logistic regression analyses were applied to vet the relationships between high blood lipid levels and prevalence of periodontal disease. Model I was analyzed on the basis of demographic characteristics, model II was analyzed on the basis of total cholesterol level and triglyceride level, and model III, was analyzed on the basis of demographic characteristics, total cholesterol level, and triglyceride level. Statistical analyses were conducted using SPSS ver. 21.0 software (IBM Co., Armonk, NY, USA), and statistical significance was determined for *p* < 0.05 (two-tailed).

## 3. Results

### 3.1. Demographic Characteristics by Total Cholesterol

With regard to gender, the proportion of females in the HTCG was the highest. With regard to age, the proportion of 50- to 69-year-olds in the HTCG was the highest. With regard to BMI, the proportion of obesity in the HTCG was the highest. With regard to marital status and education level, the proportions of married people and high-school graduates in HTCG were the highest. With regard to home ownership, number of persons living together, and health insurance coverage, the proportions in the HTCG were similar. With regard to household income and economic activity, the proportions of middle-income people and those with office jobs in the HTCG were the highest. Gender, age, BMI, and marital status showed significant differences (*p* < 0.001) (Table 1).

### 3.2. Demographic Characteristics by Triglyceride Level

With regard to gender, the proportion of males in the HTG was the highest. With regard to age, the proportion of 50- to 69-year-olds in the HTG was the highest. With regard to BMI, the proportion of obesity in the HTG was the highest. With regard to marital status and education level, the proportions of married people and high-school graduates in the HTG were the highest. With regard to home ownership, the proportion of people who did not own a house in the HTG was the highest. With regard to the number of persons living together, the proportion of those living alone in the HTG was the highest. With regard to health insurance coverage, the proportion of those with medical care in the HTG was the highest. With regard to household income, the proportion of those with a low income in the HTG was the highest. With regard to economic activity, the proportion of those with nonclerical jobs in the HTG was the highest. There were significant differences in terms of all criteria (gender, age, BMI, marital status, education level, home ownership, number of persons living together, health insurance coverage, household income, and economic activity) (Table 2).

### 3.3. CPI by Blood Lipid Levels

The CPI of the maxillary right posterior was 1.314-fold higher, the CPI of the maxillary incisor was 1.356-fold higher, the CPI of the maxillary left posterior was 0.804-fold lower, the CPI of the mandibular right posterior was 0.848-fold lower, the CPI of the mandibular incisor was 1.247-fold higher, and the CPI of the mandibular left posterior was 1.096-fold higher in the HTCG. There were no significant differences (Table 3).

The CPI of the maxillary right posterior was 0.987-fold lower, the CPI of the maxillary incisor was 1.631-fold higher, the CPI of the maxillary left posterior was 1.258-fold higher, the CPI of the mandibular right posterior was 0.970-fold lower, the CPI of the mandibular incisor was 1.575-fold higher, and the CPI of the mandibular left posterior was 0.895-fold lower in the HTG. The CPIs of the maxillary incisor and mandibular incisor in the HTG were significantly higher than those in the NTG (Table 4).

### 3.4. Relationships of the Demographic Characteristics and Blood Lipid Levels with the Prevalence of Periodontal Disease

Table 5 demonstrates the adjusted odds ratios and 95% confidence intervals from multivariable logistic regression analyses regarding the relationships of demographic characteristics and blood lipid levels with the prevalence of periodontal disease. Model I was analyzed regarding demographic characteristics with respect to the prevalence of periodontal disease. The odds ratios and 95% confidence intervals of the prevalence of periodontal disease were 2.198 (1.982–2.438) in males, 1.045 (1.010–1.050) as age increased, 1.053 (1.038–1.069) as BMI increased, 1.937 (1.613–2.323) when education level involved elementary-school graduates compared with postgraduates, and 1.587 (1.378–1.828) among those with nonclerical jobs compared with those without any economic activity.

Model II was analyzed regarding blood lipid levels with respect to the prevalence of periodontal disease. The odds ratios and 95% confidence intervals of prevalence of periodontal disease were 0.968 (0.827–1.133) in the HTCG and 1.768 (1.576–1.983) in the HTG, respectively.

Model III was analyzed regarding demographic characteristics and blood lipid levels with respect to the prevalence of periodontal disease. The odds ratios and 95% confidence intervals of the prevalence of periodontal disease were 1.834 (1.652–2.036) in males, 1.046 (1.040–1.051) as age increased, 1.047 (1.031–1.063) as BMI increased, and 1.636 (1.329–2.015) when education level involved middle-school graduates compared with postgraduates. The odds ratios and 95% confidence intervals of the prevalence of periodontal disease were 0.857 (0.742–0.954) when subjects owned a house, 1.222 (1.033–1.447) in the middle-to-low-income group compared with the high-income group, and 1.560 (1.349–1.804) among those with nonclerical jobs compared with those without any economic activity. The odds ratios and 95% confidence intervals of the prevalence of periodontal disease were 1.058 (0.897–1.247) in the HTCG and 1.499 (1.320–1.703) in the HTG. The explanatory power of model III was higher than that of model I and model II (Table 5).

## 4. Discussion

Today’s improved socioeconomic and healthcare levels have led to a keen interest in health and quality of life. Health is regarded as a comprehensive concept, and a completely healthy state is evaluated to be achieved when oral health, which is part of general health, is ensured [11]. In the early stages of periodontal disease, there are no subjective symptoms; thus, it is easy for it to progress chronically, and, as a result, tooth loss, lowered masticatory ability, lack of nutrition, and poor quality of life appear [12]. Periodontal disease affects 30–35% of the global adult population, and it is reported that its prevalence is gradually increasing [13]. In particular, an association between cardiovascular and periodontal diseases was recently reported.

According to a study by Emingil et al. [14], a high bleeding frequency and many ≥4 mm periodontal pockets appear during periodontal probing in patients with a history of acute myocardial infarction. A study on the relationship between prognosis before and after treatment of coronary artery disease reported that the periodontal health status examined during the 7 year follow-up period was associated with coronary artery disease [15].

Among the major factors influencing periodontal disease, systemic disease factors are continuously being studied [16]. Low blood high-density-lipoprotein (HDL) levels were reported to be a risk factor for cardiovascular disease in many previous epidemiological studies [17]. In addition, Choi et al. [18], Nepomuceno et al. [19], and Abraham et al. [20] reported that triglyceride level, HDL-cholesterol(HDL-C), and low HDL-C, respectively, are significantly associated with periodontitis.

As dyslipidemia is related to social, political, and economic variables, it is necessary to verify the blood lipid levels according to the demographic characteristics. In this study, males and 50- to 69- year-olds had high total cholesterol level. In terms of marital status, married people had high total cholesterol level, consistent with the result of the study by Gordon et al. [17]. Remarkably, subjects with high triglyceride level showed significant differences in terms of all the variants such as gender, age, marital status, education level, home ownership, numbers of persons living together, health insurance coverage, household income, and economic activity. In particular, the proportion of subjects with nonclerical jobs in the high-triglyceride group was higher than that of subjects with office jobs. This is in agreement with the result of the study by Jaramillo et al. [21], which showed that 80.2% of the subjects who had advanced periodontitis were in the low socioeconomic level.

The results of this study showed that hyperlipidemia is connected with periodontal disease. In terms of total cholesterol level, the group with a high total cholesterol level tended to have more periodontal disease expression than the group with a normal total cholesterol level, but the difference between both groups was not statistically significant. This is consistent with the study result obtained by Machado et al. that periodontal disease was not associated with blood concentration because there was no difference in the prevalence of periodontitis between the group with high total cholesterol and triglyceride levels and the group with low total cholesterol and triglyceride levels [22]. The same result was obtained in the study by Banihashemrad et al. Although there were direct correlations between the prevalence of periodontal disease and the measured biochemical parameters, no correlations were found to be significant [23]. Other studies reported, however, that a higher total cholesterol level leads to a greater prevalence of periodontal disease [18,24,25,26,27].

In terms of triglyceride level, the expression level of periodontal disease in people with a high triglyceride level was 1.499-fold higher than in those whose triglyceride levels were within the normal range, and the difference was statistically significant. Even after correction with various demographic variables, such as gender, age, education level, home ownership, household income, and economic activity, the value was significant, and the confidence interval was 1.320–1.703. The results of this study are the same as those of the study by Shivakumar et al. [27], who conducted their study on people with dyslipidemia and a control group. The hyperlipidemic patients showed significantly higher values across the four periodontal parameters. Plasma triglyceride, total cholesterol, and low-density-lipoprotein cholesterol (LDL-C) were significantly and positively associated with probing depth and bleeding on probing [27]. In addition, the results of the study by Pejcic, which investigated the reduction of blood lipid levels after periodontal treatment in patients with periodontitis and the control group, also showed high blood concentrations of triglyceride in patients with periodontitis before periodontal treatment [28]. In many other studies, a high triglyceride level was associated with a high prevalence of periodontal disease [18,24,25,26,29]. As triglyceride levels increased, the severity of periodontal disease also increased [22,30,31,32,33]. This means that high blood lipid levels may be a risk factor of periodontal disease. Therefore, people with progressive periodontal disease should control their triglyceride levels.

As data from KNHANES VII were analyzed in this study, this result may be generalized to the adult population of Korea aged above 19 years. However, this is a cross sectional study; thus, it may not be able to predict disorder. Although the results support the association between periodontal disease and hypertriglyceridemia, more studies with follow-up cohort designs need to be carried out to clarify the actual relationship between high blood lipid level and periodontal condition.

## 5. Conclusions

This study confirmed that a high triglyceride level certainly has a negative effect on periodontal health. However, a high total cholesterol level is not related to the prevalence of periodontal disease in South Koreans. In conclusion, management of triglyceride level could be a possible precaution of periodontal disease. Thus, people who want to maintain a healthy periodontal status should control their triglyceride levels.

## Figures and Tables

**Table 1 healthcare-08-00337-t001:** Demographic characteristics by total cholesterol level.

Characteristics	Sub-Items	NTCG	HTCG	*p*
Gender	Male	5790 (91.6)	547 (8.4)	0.000 *
	Female	6943 (89.6)	788 (10.4)	
	Total	12,733 (90.4)	1335 (9.6)	
Age (years)	<30	2755 (96.4)	92 (3.6)	0.000 *
	30–49	3949 (89.3)	483 (10.7)	
	50–69	4015 (86.8)	601 (13.2)	
	≥70	2014 (93.0)	159 (7.0)	
	Total	12,733 (90.4)	1335 (9.6)	
BMI (kg/m^2^)	Underweight (<18.5)	840 (96.4)	28 (3.6)	0.000 *
	Normal (18.5–24.9)	7720 (91.5)	720 (8.5)	
	Obese (≥25.0)	4020 (87.2)	569 (12.8))	
	Total	12,580 (90.4)	1317 (9.6)	
Marital status	Married	7822 (89.2)	963 (10.8)	0.000 *
	Single	4908 (92.6)	372 (7.4)	
	Total	12,730 (90.4)	1335 (9.6)	
Education level	<Elementary	2030 (90.0)	226 (10.0)	0.190
	Middle school	1607 (91.8)	134 (8.2)	
	High school	3291 (89.8)	377 (10.2)	
	>College	5008 (90.3)	543 (9.7)	
	Total	11,936 (90.3)	1280 (9.7)	
Home ownership	No	3843 (90.5)	409 (9.5)	0.808
	Yes	8890 (90.4)	926 (9.6)	
	Total	12,733 (90.4)	1335 (9.6)	
Number of persons living together	Alone	1466 (89.2)	170 (10.8)	0.164
	>2 persons	11,267 (90.6)	1165 (9.4)	
	Total	12,733 (90.4)	1335 (9.6)	
Health insurance coverage	National Health Insurance	12,152 (90.3)	1288 (9.7)	0.230
	Medical care	486 (92.9)	38 (7.1)	
	Total	12,638 (90.4)	1326 (9.6)	
Household income	Low	2077 (90.9)	218 (9.1)	0.291
	Middle–low	2372 (91.3)	224 (8.7)	
	Middle	2589 (89.6)	288 (10.4)	
	Middle–high	2838 (90.8)	283 (9.2)	
	High	2823 (89.9)	318 (10.1)	
	Total	12,699 (90.4)	1331 (9.6)	
Economic activity	None	4733 (90.4)	482 (9.6)	0.233
	Office job	4208 (89.4)	512 (10.6)	
	Nonclerical job	2525 (90.5)	279 (9.5)	
	Total	11,466 (90.0)	1273 (10.0)	

BMI (body mass index), NTCG (normal-total-cholesterol group), HTCG (high-total-cholesterol group); * *p* < 0.05.

**Table 2 healthcare-08-00337-t002:** Demographic characteristics by triglyceride level.

Characteristics	Sub-Items	NTG	HTG	*p*
Gender	Male	5085 (80.6)	1252 (19.4)	0.000 *
	Female	6985 (90.3)	746 (9.7)	
	Total	12,070 (86.0)	1998 (14.0)	
Age (years)	<30	2682 (94.1)	165 (5.9)	0.000 *
	30–49	3691 (83.9)	741 (16.1)	
	50–69	3808 (82.9)	808 (17.1)	
	≥70	1889 (87.1)	284 (12.9)	
	Total	12,070 (86.0)	1998 (14.0)	
BMI (kg/m^2^)	Underweight (<18.5)	854 (98.5)	14 (1.5)	0.000 *
	Normal (18.5–24.9)	7576 (90.3)	864 (9.7)	
	Obese (≥25.0)	3499 (75.8)	1090 (24.2)	
	Total	11,929 (96.1)	1968 (13.9)	
Marital status	Married	7403 (84.8)	1382 (15.2)	0.000 *
	Single	4664 (88.2)	616 (11.8)	
	Total	12,067 (86.0)	1998 (14.0)	
Education level	<Elementary	1913 (84.8)	343 (15.2)	0.003 *
	Middle school	1507 (86.2)	234 (13.8)	
	High school	3085 (84.7)	583 (15.3)	
	>College	4822 (87.4)	729 (12.6)	
	Total	11,327	1889	
Home ownership	No	3922 (84.4)	740 (15.6)	0.012 *
	Yes	3588 (84.8)	664 (15.2)	
	Total	8482 (86.6)	1334 (13.4)	
Number of persons living together	Alone	1367 (84.0)	269 (16.0)	0.026 *
	>2 persons	10,703 (86.3)	1729 (13.7)	
	Total	12,070 (86.0)	1998 (14.0)	
Health insurance coverage	National Health Insurance	11,557 (86.2)	1883 (13.8)	0.035 *
	Medical care	430 (82.6)	94 (17.4)	
	Total	11,987 (86.1)	1977 (13.9)	
Household income	Low	2864 (84.1)	560 (15.9)	0.025 *
	Middle–low	2051 (85.2)	345 (14.8)	
	Middle	1829 (86.7)	300 (13.3)	
	Middle–high	1624 (86.5)	237 (13.5)	
	High	3639 (87.4)	547 (12.6)	
	Total	12,007 (86.0)	1989 (14.0)	
Economic activity	None	4570 (87.8)	645 (12.2)	0.000 *
	Office job	4005 (85.3)	715 (14.7)	
	Nonclerical job	2293 (81.9)	511 (18.1)	
	Total	10,868 (85.7)	1871 (14.3)	

NTG (normal-triglyceride group), HTG (high-triglyceride group); * *p* < 0.05.

**Table 3 healthcare-08-00337-t003:** Community periodontal index (CPI) by total cholesterol level.

Characteristics	NTCG	HTCG		
	OR	*p*	OR	95% CI
Maxillary right posterior	1	0.481	1.314	0.614–2.813
Maxillary incisor	1	0.249	1.356	0.808–2.275
Maxillary left posterior	1	0.535	0.804	0.403–1.605
Mandibular right posterior	1	0.472	0.848	0.541–1.330
Mandibular incisor	1	0.094	1.247	0.963–1.614
Mandibular left posterior	1	0.713	1.096	0.671–1.792

OR (odds ratio), CI (confidence interval), NTCG (normal-total-cholesterol group), HTCG (high-total-cholesterol group).

**Table 4 healthcare-08-00337-t004:** CPI by triglyceride level.

Characteristics	NTG	HTG		
	OR	*p*	OR	95% CI
Maxillary right posterior	1	0.963	0.987	0.563–1.731
Maxillary incisor	1	0.028 *	1.631	1.055–2.520
Maxillary left posterior	1	0.279	1.258	0.830–1.907
Mandibular right posterior	1	0.849	0.970	0.708–1.319
Mandibular incisor	1	0.000 *	1.575	1.290–1.922
Mandibular left posterior	1	0.562	0.895	0.616–1.301

NTG (normal-triglyceride group), HTG (high-triglyceride group); * *p* < 0.05.

**Table 5 healthcare-08-00337-t005:** The odds ratios and 95% confidence intervals of demographic characteristics and blood lipid levels with respect to the prevalence of periodontal disease.

Characteristics	Sub-Items	Model I			Model II			Model III		
		*p*	OR	95% CI	*p*	OR	95% CI	*p*	OR	95% CI
Gender	Male	0.000 *	2.198	1.982–2.438				0.000 *	1.834	1.652–2.036
	Female		1						1	
Age (years)		0.000 *	1.045	1.010–1.050				0.000 *	1.046	1.040–1.051
BMI (kg/m^2^)		0.000 *	1.053	1.038–1.069				0.000 *	1.047	1.031–1.063
Marital status	Married	0.092	1.111	0.983–1.257				0.059	1.148	0.995–1.325
	Single		1						1	
Education level	<Elementary	0.000 *	1.937	1.613–2.3235				0.000 *	1.620	1.331–1.971
	Middle school	0.000 *	1.765	1.447–2.152				0.000 *	1.636	1.329–2.015
	High school	0.000 *	1.559	1.355–1.794				0.000 *	1.563	1.359–1.798
	>College		1						1	
Home ownership	Yes	0.002 *	0.811	0.718–0.916				0.043 *	0.857	0.742–0.954
	No		1						1	
Number of persons living together	Alone	0.576	0.864	0.720–1.036				0.969	0.974	0.809–1.174
	>2 persons		1						1	
Health insurance coverage	National Health Insurance	0.115	1.084	0.828–1.418				0.555	0.892	0.671–1.185
	Medical care		1						1	
Household income	Low	0.147	1.174	0.945–1.459				0.096	1.149	0.939–1.406
	Middle–low	0.188	1.133	0.941–1.364				0.039 *	1.222	1.033–1.447
	Middle	0.223	1.113	0.936–1.324				0.175	0.969	0.816–1.151
	Middle–high	0.651	0.965	0.825–1.128				0.651	0.981	0.820–1.174
	High		1						1	
Economic activity	Nonclerical job	0.000 *	1.587	1.378–1.828				0.000 *	1.560	1.349–1.804
	Office job	0.005 *	1.315	1.155–1.496				0.015 *	1.324	1.157–1.515
	None		1						1	
Total cholesterol	HTCG				0.689	0.968	0.827–1.133	0.590	1.058	0.897–1.247
	NTCG					1			1	
Triglyceride	HTG				0.000 *	1.768	1.576–1.983	0.000 *	1.499	1.320–1.703
	NTG					1			1	

Data were obtained through multivariable logistic regression analysis using complex-sample design; NTCG (normal-total-cholesterol group), HTCG (high-total-cholesterol group), NTG (normal-triglyceride group), HTG (high-triglyceride group); * (*p* < 0.05). Model I: Nagelkerke *R^2^* = 0.214, *p* < 0.001; Model II: Nagelkerke *R^2^* = 0.013, *p* < 0.001; Model III: Nagelkerke *R^2^* = 0.218, *p* < 0.001.

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
