# Peer review of "Association between Periodontal Disease and Levels of Triglyceride and Total Cholesterol among Korean Adults"

_healthcare, 2020, doi:10.3390/healthcare8030337_

Round 1
Reviewer 1 Report
Introduction
There are a lot of redundancies in the introduction, the manuscript need to be revised by a native english speaker in order to make it more readable.
In the introduction I suggest to cite this paper: DOI: https://doi.org/10.1152/ajpgi.00161.2019 - PMID: 32146836. There is a section on "the impact of oral bacteria", which explains how ingestion of pathogen involved in periodontitis can lead to liver disease.
Methods
Line 63: "was consisted" is wrong, change in "consisted".
Line 64: "cress sectional", change in "cross".
Line 80: The authors define income into some category, can they define numerically the incomes?
Statistical analysis: the Chi-Square test does not study "relationships", but Inter group differences. Also, was the p < 0.05 two-tailed? Please clarify.
Also, how was the multivariate model performed? (step-wise forward, backward, etc?) And how was the best model chosen?
There is an important limitation: why did the author not evaluate weight and height (BMI) that could influence blood cholesterol?
Author Response
Introduction
There are a lot of redundancies in the introduction, the manuscript need to be revised by a native english speaker in order to make it more readable.
In the introduction I suggest to cite this paper: DOI: https://doi.org/10.1152/ajpgi.00161.2019 - PMID: 32146836. There is a section on "the impact of oral bacteria", which explains how ingestion of pathogen involved in periodontitis can lead to liver disease.
→ We added the paper: DOI: https://doi.org/10.1152/ajpgi.00161.2019 - PMID: 32146836
Methods
Line 63: "was consisted" is wrong, change in "consisted".
→ We changed ""was consisted" to "consisted".
Line 64: "cress sectional", change in "cross".
→ We changed "cress” to "cross".
Line 80: The authors define income into some category, can they define numerically the incomes?
→The household income quintile, an indicator used in the National Health and Nutrition Survey, was used as it was.
Statistical analysis: the Chi-Square test does not study "relationships", but Inter group differences.
→ We changed "relationships” to “differences”
Also, was the p < 0.05 two-tailed? Please clarify.
→ We edited ‘the p < 0.05 two-tailed’
Also, how was the multivariate model performed? (step-wise forward, backward, etc?)
→ Complex-samples Logistic regression analysis that analyzes the factors affecting the prevalence of periodontal disease is not a step-wise check.
And how was the best model chosen?
→ The best model was model III as shown in table 5.
There is an important limitation: why did the author not evaluate weight and height (BMI) that could influence blood cholesterol?
→ We added BMI.
Reviewer 2 Report
In the study entitled “Association between periodontal disease and levels of triglyceride and total cholesterol among Korean adults” the authors investigated the causal relationship between periodontal and cardiovascular diseases and to provide fundamental data suggesting the importance of periodontal care by analyzing the impact of the presence or absence of dyslipidemia on the induction of periodontal disease using data from the 7th Korea National Health and Nutrition Examination Survey (KNHANES â…¦) conducted for 3 years, from 2016 to 2018. They further hypothesized, that with the increasing life expectancy, the importance of major oral diseases that threaten oral health has been emphasized more of late. As the desire for good health has increased, the interest in oral health, an essential element of health, has also increased. Periodontal disease occurs in the surrounding tissues supporting the teeth, and most periodontal diseases take the form of chronic inflammatory periodontitis, that progresses slowly but progresses and stops repeatedly depending on the individual’s systemic health and oral care status. Periodontal disease is a major oral health problem and causes loss of teeth by destroying the connective tissue and bones that support the teeth. Consequentially the authors have focused on the relationship between periodontal disease and cardiovascular disease, a mediator that causes inflammation in the oral cavity affects the inflammatory response of the periodontal tissue and the whole body through the bloodstream. In addition, it has been reported that various risk factors, such as health behaviors and daily lifestyle habits, have a common effect on periodontal and cardiovascular diseases. Previous studies have reported that cardiovascular disease and periodontal disease are connected to each other. The studies on the relationship between dyslipidemia and periodontal disease, however, are very insufficient. Due to the changes in eating habits and lifestyles with the rapid advance of the modern society, the trend of disease occurrence also changes, so it must be continuously managed based on the relationship between systemic and oral diseases. Therefore, much attention and effort are needed to promote oral health.
Indeed, this approach is not complete novel. In a large study (Choi YH, Kosaka T, Ojima M, et al. BMC Oral Health. 2018;18(1):77. Published 2018 May 4. doi:10.1186/s12903-018-0536-0) the authors investigated the „Relationship between the burden of major periodontal bacteria and serum lipid profile in a cross-sectional Japanese study”. Furthermore, Nepomuceno R, Pigossi SC, Finoti LS, et al. reported about the “Serum lipid levels in patients with periodontal disease: A meta-analysis and meta-regression” in the J Clin Periodontol. 2017;44(12):1192-1207. doi:10.1111/jcpe.12792. Several papers have considered the potential relationship between periodontitis and lipid parameters. Eligible studies were those with data about serum lipid parameter levels in non-smoking subjects with and without chronic periodontitis, who are generally healthy and not taking any medication for dyslipidemia. Mean differences and 95% confidence intervals for total cholesterol, triglycerides, low-density lipoprotein (LDL) cholesterol and high-density lipoprotein (HDL) cholesterol were obtained from all the selected studies. Participants with chronic periodontitis presented significantly higher serum levels of LDL and triglycerides (p = .003 and p < .0001, respectively). The total cholesterol was higher in the PD group, but without significant difference in comparison with healthy participants. Significantly (p = .0005) lower HDL serum levels were found in patients with chronic periodontitis than in healthy subjects.
The cited authors came to the same conclusions as the authors of reviewed publication, it is suggested that PD is significantly associated with reduction in HDL and elevation of LDL and triglyceride concentrations. This analysis supports the rationale that periodontal disease is associated with lipid metabolic control. Notably, no one of these studies is cited in the reviewed publication. Obviously, the authors are not aware of the current literature in the field (too many to cite them all, but a quick pubmed search should be sufficient).
Author Response
In the study entitled “Association between periodontal disease and levels of triglyceride and total cholesterol among Korean adults” the authors investigated the causal relationship between periodontal and cardiovascular diseases and to provide fundamental data suggesting the importance of periodontal care by analyzing the impact of the presence or absence of dyslipidemia on the induction of periodontal disease using data from the 7th Korea National Health and Nutrition Examination Survey (KNHANES â…¦) conducted for 3 years, from 2016 to 2018. They further hypothesized, that with the increasing life expectancy, the importance of major oral diseases that threaten oral health has been emphasized more of late. As the desire for good health has increased, the interest in oral health, an essential element of health, has also increased. Periodontal disease occurs in the surrounding tissues supporting the teeth, and most periodontal diseases take the form of chronic inflammatory periodontitis, that progresses slowly but progresses and stops repeatedly depending on the individual’s systemic health and oral care status. Periodontal disease is a major oral health problem and causes loss of teeth by destroying the connective tissue and bones that support the teeth. Consequentially the authors have focused on the relationship between periodontal disease and cardiovascular disease, a mediator that causes inflammation in the oral cavity affects the inflammatory response of the periodontal tissue and the whole body through the bloodstream. In addition, it has been reported that various risk factors, such as health behaviors and daily lifestyle habits, have a common effect on periodontal and cardiovascular diseases. Previous studies have reported that cardiovascular disease and periodontal disease are connected to each other. The studies on the relationship between dyslipidemia and periodontal disease, however, are very insufficient. Due to the changes in eating habits and lifestyles with the rapid advance of the modern society, the trend of disease occurrence also changes, so it must be continuously managed based on the relationship between systemic and oral diseases. Therefore, much attention and effort are needed to promote oral health.
Indeed, this approach is not complete novel. In a large study (Choi YH, Kosaka T, Ojima M, et al. BMC Oral Health. 2018;18(1):77. Published 2018 May 4. doi:10.1186/s12903-018-0536-0) the authors investigated the „Relationship between the burden of major periodontal bacteria and serum lipid profile in a cross-sectional Japanese study”. Furthermore, Nepomuceno R, Pigossi SC, Finoti LS, et al. reported about the “Serum lipid levels in patients with periodontal disease: A meta-analysis and meta-regression” in the J Clin Periodontol. 2017;44(12):1192-1207. doi:10.1111/jcpe.12792. Several papers have considered the potential relationship between periodontitis and lipid parameters. Eligible studies were those with data about serum lipid parameter levels in non-smoking subjects with and without chronic periodontitis, who are generally healthy and not taking any medication for dyslipidemia. Mean differences and 95% confidence intervals for total cholesterol, triglycerides, low-density lipoprotein (LDL) cholesterol and high-density lipoprotein (HDL) cholesterol were obtained from all the selected studies. Participants with chronic periodontitis presented significantly higher serum levels of LDL and triglycerides (p = .003 and p < .0001, respectively). The total cholesterol was higher in the PD group, but without significant difference in comparison with healthy participants. Significantly (p = .0005) lower HDL serum levels were found in patients with chronic periodontitis than in healthy subjects.
→ We added two papers.
- Choi YH, et al. BMC Oral Health. 2018;18(1):77. doi:10.1186/s12903-018-0536-0. Relationship between the burden of major periodontal bacteria and serum lipid profile in a cross-sectional Japanese study.
- Nepomuceno R, et al. doi:10.1111/jcpe.12792. “Serum lipid levels in patients with periodontal disease: A meta-analysis and meta-regression” J Clin Periodontol. 2017;44(12):1192-1207.
The cited authors came to the same conclusions as the authors of reviewed publication, it is suggested that PD is significantly associated with reduction in HDL and elevation of LDL and triglyceride concentrations. This analysis supports the rationale that periodontal disease is associated with lipid metabolic control. Notably, no one of these studies is cited in the reviewed publication. Obviously, the authors are not aware of the current literature in the field (too many to cite them all, but a quick pubmed search should be sufficient.
→ There are still different conclusions between blood lipids and periodontal disease, so we thought additional research are needed to clarify the association between blood lipids and periodontal disease . The consequences of what has been dealt with as risk factors are also inconsistent. In particular, the results of studies on large numbers of people, not clinical trials, are even more different. Therefore, we think it is meaningful because our research is based on the results of the National Health and Nutrition Survey, which was conducted on systemic samples throughout the nation for three years.
Round 2
Reviewer 1 Report
authors have addressed reviewer's concerns,
the manuscript is ready for publication.
Reviewer 2 Report
I agree with the additional provided informations.